# Autism Spectrum Disorder as a Multifactorial Disorder: The Interplay of Genetic Factors and Inflammation

**DOI:** 10.3390/ijms26136483

**Published:** 2025-07-05

**Authors:** George Ayoub

**Affiliations:** Psychology, Santa Barbara City College, Santa Barbara, CA 93109, USA; neuro@sbcc.edu

**Keywords:** autism, ASD, leucovorin, folate, cerebral folate deficiency, inflammation, oxidative stress, maternal immune activation

## Abstract

Autism spectrum disorder (ASD) is a complex neurodevelopmental condition characterized by difficulty with social communication, behavior, and sensory integration. With its prevalence rising worldwide in recent decades, understanding and mitigating the origins of ASD has become a priority. Though its etiology is multifactorial, the current research highlights two major contributors, genetic susceptibilities and environmental inflammatory exposures, leading to oxidative stress during critical developmental periods. We explore how genetic variations, including those affecting cerebral folate metabolism, and various inflammatory triggers, including exposure to inflammatory agents during both the fetal and post-fetal period, intersect to influence the development of ASD, giving rise to specific symptoms seen in autism.

## 1. Introduction

Autism spectrum disorder (ASD) is a developmental disorder identified in the American Psychiatric Association’s Diagnostic and Statistical Manual, Fifth Edition (DSM-5), by two categories of symptoms: persistent deficits in social communication and social interaction, and restricted, repetitive patterns of behavior. The DSM-5 further provides guidelines as to severity, listing three levels. Level 1 requires support and may struggle with initiating social interactions. Level 2 requires substantial support and may display limited communication. Level 3 requires very substantial support and shows significant challenges in communication and behavior [1].

The diagnosis of ASD has seen a dramatic rise in the 21st century, increasing approximately five-fold and becoming a public health priority [2]. The Simons Foundation created the autism research initiative (SFARI), which houses the evolving database of the genetics of autism [3,4]. There are striking parallels between ASD and other multifactorial diseases such as cancer. In both conditions, risk is influenced by an interplay of genetic predispositions and environmental factors, particularly triggers of inflammation, which lead to oxidative stress. Just as cancer risk can be increased by inherited mutations and external exposures like toxins or infections, ASD may also arise from a combination of genetic vulnerability and environmental stressors. In both diseases, these factors modulate susceptibility rather than determining certainty of onset, effectively raising the probability of development becoming neurodivergent.

Furthermore, recent findings in both fields suggest an important role of the gut microbiome in modulating immune and neurological outcomes. Nutritional and dietary choices appear to influence disease trajectories in both ASD and cancer. For example, diets that reduce inflammation or support detoxification pathways are known to reduce the risk for cancer [5] and depression [6], and may point a path to better outcomes in ASD with nutritional intervention as well.

Another similarity with cancer is that ASD is presumed to have multiple variants with differing etiologies. To date, over 70 genes have been correlated with ASD, with hundreds more believed to be related to ASD susceptibility [7,8,9]. This wide array of genes may help explain the diversity of expression of ASD, as well as point to a rationale for some of the comorbidities commonly diagnosed with ASD [10].

These insights invite a new perspective on ASD not just as a static diagnosis but as a dynamic condition potentially influenced by modifiable risk factors. This opens the door to discussing strategies for reducing risk, and even the possibility of partial or complete remission under certain therapeutic regimens. Understanding the contributing causes of ASD could lead to more personalized, preventive, and responsive treatments.

## 2. Genetic Factors in ASD

Many hundreds of genes have been implicated as having a role in autism, with multiple developmental impact categories involved [11,12,13]. Comparisons between the AutDB, AutismKB2.0, and SFARI Gene databases show over 400 genes presumed to be linked to ASD [14]. Previously, categorizing autism as syndromic or non-syndromic was useful to obtain a sense of some genes involved, especially for monogenic disorders [15]. The use of such terminology is considered by some to be limiting now [14]. Here, we focus on non-syndromic autism, take a look at some of the more frequently referenced ASD-related genes, and consider the categories of developmental impact these may have to understand which of these categories may be impacted.

We note the substantial positive work in monogenic syndromic autism, where gene therapies can be developed to target the single genetic cause [16,17]. The benefits of this work cannot be overestimated. For the purposes of this review, we seek to draw from the non-syndromic ASD evidence to begin to obtain a sense of the many factors involved in the majority of autism cases.

The following section will look at factors of oxidative stress to provide a compendium of the inflammatory agents that may influence gene expression during critical developmental periods. We hope to provide a sufficient overview of the genes involved to give context for putative inflammation-induced oxidative stress actions, with the goal of stimulating further research in the synergism of oxidative stress on gene regulation and expression in the development of autism.

### 2.1. Categorizing ASD Genes

Kereszturi (2024) [18] compiled the most commonly referenced genes listed in three databases listing autism-related genes: SFARI, AutDB, and ClinVar. These were then compared for their functional impact and listed in ten areas of developmental differences, which are summarized in the following table. The categories fall into three developmental areas, as shown in Table 1: synaptic, social, and neuronal.

One challenge here is that not only are some genes involved in multiple aspects of development but there are some that are apparently involved in quite different categories of action. For example, *SHANK3* is believed to play a role in morphogenesis, synaptic organization, and social behavior. Given that the *SHANK1*, *SHANK2*, and *SHANK3* genes produce proteins that are differentially expressed in development and affect synaptic structure and neurotransmission, this is unsurprising [19]. However, the multiplicity of categories makes the matching of specific genes to specific traits difficult at best, and potentially misleading.

The gene *UBE3A* codes for ubiquitin ligase (and its coactivator). The loss of this gene activity causes Angelman syndrome, with lack of speech, seizures, autistic features, and intellectual disability. The gain of function mutations of *UBE3A*, as well as duplication, also results in autism behaviors [20]. It is the impact of *UBE3A* on synaptic function and plasticity that appears to be the causative factor [20]. There are other genes involved in ubiquitination, and recent work indicates that these also create autism-like symptoms in an animal model, as with CUL3 [21], or are directly involved in human autism [22]. These demonstrate the critical role of ubiquitination in neuronal synapses as regards autism.

Other sources have evaluated genes based on pathway participation and functional clustering in ASD [23,24,25]. This allowed us to construct Table 2 to compare the list of genes with the pathways involved.

The genes that have been recognized as ASD-related and act by impairing neural connectivity include those involved in synapse formation and function, namely *SHANK3*, *NRXN1*, and *NGLG3/4* [26,27,28]. Additionally, the gene *SCN2A* is tied to an alteration in neuronal activity due to its role in ion channels [29].

The genes involved in cell growth, *TSC1* and *TSC2*, can lead to abnormal brain structures by alterations in the mTOR pathway [30,31]. This is a common signaling cascade with central roles in translation, lipid and nucleotide synthesis, and growth factor signaling [32].

Variations in the genes involved in gene regulation, *CHD8* and *MECP2*, can result in disrupted brain development, while variants of a gene from RNA translation, *FMR1*, is related to learning and memory issues [33,34,35].

Genes associated with folate transport (Section 2.3) are related to folate delivery to the brain and are a biomarker for cerebral folate deficiency (CFD). Therapeutic intervention can overcome CFD, as described in this section. A potentially related genetic risk is for mitochondrial dysfunction, which may play a role in the autism regression seen in some children as they grow.

### 2.2. Comorbidities of ASD-Related Genes

Another route is to evaluate the disorders comorbid with ASD and which of the ASD-associated genes are also implicated in the comorbid disorder. In a compelling analysis of SFARI data, Khachadourian et al. (2023) [10] identified many comorbidities found with ASD. Such data may be doubly beneficial. There is the main point of helping to identify the specific genes involved in certain developmental symptoms. But the second benefit may come later in providing a means to begin to characterize the spectrum of autism as specific subtypes, based either on the comorbidities or, hopefully, on a full understanding of the genetic and environmental mechanisms involved in the development of these symptoms.

In addition to comorbidities with learning disorders and social communication disorders, some of the common comorbidities found were with anxiety disorder, ADHD, depression, dystonia (motor disorders), and OCD [10,36]. Table 3 compiles the more frequently referenced SFARI-listed genes that are linked to one of the listed comorbidities [37,38].

The psychiatric cell map initiative [39] is an integrated effort to identify genes that are believed to be involved in multiple disorders and evaluate their relationship [40]. It shows ASD having comorbidities with certain epileptic genes, *GABRB3* and *SCN2A*, as well as intellectual disability genes, *SCN2A*, *SLC6A1*, *SYNGAP1*, and *WAC* [39]. This work was taken further to make a connected cluster map to the curated pathways of SFARI proteins and their neurological relevance and reveal the overlap of neurodevelopmental disease causal networks (in their Figure 6) [41].

Additionally, there is evidence from brain organoids of a form of idiopathic autism in which some of the genes involved lead to an imbalance in excitatory cortical neurons [42]. Given that it is presumed that there are several autism variants, the associated genes are ones that are correlated with a greater risk of developing ASD. Some have specific pathways that are involved, and others are correlated with ASD but in a non-defined manner. Section 2.3 and Section 2.4 provide a brief look at some of the developmental genes that have traditionally been considered for their role in ASD.

### 2.3. Cerebral Folate Deficiency

#### 2.3.1. Folate Metabolism and Brain Development 

Folate is essential for DNA synthesis, repair, and methylation, all of which are essential processes during early brain development. The transport of folate into the central nervous system (CNS) is tightly regulated by specific transport mechanisms, primarily the folate receptor alpha (FRα) and the reduced folate carrier (RFC).

#### 2.3.2. Genetic Variants Impairing Folate Transport

Mutations in the *FOLR1* gene can impair FRα function, leading to CFD. Similarly, polymorphisms such as the *SLC19A1* variant rs1051266 can reduce RFC efficiency. Both impair the brain’s ability to maintain adequate folate levels, resulting in developmental impairments and ASD features, such as decreased social communication and decreased emotional recognition [43,44].

#### 2.3.3. Folate Receptor Alpha Autoantibodies and Treatments

Folate receptor autoantibodies (FRAAs), which block or bind to FRα, are much more prevalent in ASD children (about 70% of ASD children, compared with 5–10% in the general population). These autoantibodies suppress folate transport across the blood–brain barrier, contributing to CFD. FRAAs are believed to result from environmental exposure [45], further linking genetic and inflammatory pathways.

Folinic acid supplementation has shown promise in reversing CFD symptoms in children with ASD in multiple clinical trials (based in the USA, France, and India) published between 2018 and 2025 [46,47,48,49]. Dietary modifications, such as avoiding cow’s milk products (which may trigger FRAAs) [45], have also demonstrated potential benefits. These findings support the clinical importance of identifying and treating CFD in ASD patients.

### 2.4. Mitochondrial Dysfunction

There is growing evidence that mitochondrial dysfunction may contribute to neurodevelopmental regression in children [50,51,52,53,54]. This condition is seen when an ASD child shows strong learning, language, and memory capabilities initially, only to regress later in childhood [55]. Of note, while non-regressed children showed that inflammation and immune dysregulation occurs first, in regressed children, mitochondrial dysfunction was first observed, with an impact on immune activity possibly occurring later [52].

## 3. Inflammation During Critical Developmental Periods and ASD Risk

### 3.1. The Developing Brain and Vulnerability to Inflammation 

Neurodevelopment is highly sensitive to environmental insults during the prenatal and early postnatal periods. Inflammation during these times can disrupt neuronal migration, synaptogenesis, synaptic pruning, and brain connectivity, all of which are critical to neurotypical brain function [56]. Many causes of inflammation have been documented, some of which are acute, such as viral infection and fever, and others chronic, such as microplastics and malnutrition. When inflammation occurs during critical periods in brain development, neurotypical development can be delayed or arrested, resulting in a child reaching milestones later, aberrantly, or possibly not at all if the inflammation exceeds the duration of the critical period [43,56,57,58]. Summary schematics of the categories of inflammation and their presumed action are outlined in Figure 1 and Figure 2.

### 3.2. Categories of Inflammation

#### 3.2.1. Viral Infections 

Maternal infections such as influenza, rubella, and COVID-19 have been linked to increased ASD risk due to maternal immune activation (MIA). Postnatal viral infections can also contribute to neuroinflammatory states [43,57,59,60,61].

An evaluation of studies in a meta-analysis shows there is an association between maternal infection and subsequent autism diagnosis in the child, with little variation based on when the infection occurred during pregnancy or what type of infection it was [62,63]. A genetic analysis showed that the increased risk for ASD with maternal infections may be correlated with a genetically distinct subtype of autism, a subtype that come from the interaction between genetic susceptibility and the exposure to infections in pregnancy [64]. Thus, since the maternal infections may not be causal in the development of ASD, it is unclear if the prevention of infection may reduce autism incidence [65].

#### 3.2.2. Air Pollution and Maternal Asthma

Environmental pollutants are associated with ASD incidence [66]. Exposure to particulate matter (e.g., PM2.5) during pregnancy has been associated with neuroinflammation [67] and developmental disorders, and particularly an increased risk of development of ASD [56,67,68,69,70,71,72]. Airborne toxins can activate microglia and oxidative stress pathways in the fetal brain [73].

Additional studies have examined air pollution from roadways as being particularly problematic, pointing to a potential link to plasticizers used in the manufacture of tires and tailpipe exhausts, as risk was related to distance from roadways [74,75]. Given that immune dysregulation is a common comorbid feature with ASD, the link to maternal exposure to air pollution may be indicating a mechanism of action in the development of autism. In this regard, multiple studies have found an elevated risk of autism in infants born to asthmatic mothers, and that this risk was related to an increase in inflammatory biomarkers [76,77,78].

Of concern, in a study of air pollution pre-conception, women exposed to particulate matter and nitric oxide in the three months prior to conception had children with greater growth in their body weight in their first two years of life [79]. The authors report that the critical period of exposure is the three months prior to conception.

#### 3.2.3. Maternal Immune Activation

Maternal inflammation during pregnancy is associated with an increased risk of ASD, with maternal immune activation (MIA) being evaluated in animal models, where it leads to behavioral changes resembling ASD [80]. This immune activation raises the levels of cytokines, especially interleukin-6 (IL-6) and interleukin 17a (IL-17a), which are implicated in disrupting fetal brain development, in what may lead to some ASD behaviors. The effect of MIA on the developing fetus may be impacted by the fetus’s genetic makeup, with certain genetic profiles making the fetal brain more impacted by maternal inflammation, implicating a gene–environment interaction in the etiology of ASD [80,81,82,83].

#### 3.2.4. Microplastics

Emerging studies in animal models suggest that prenatal and early postnatal exposure to microplastics, especially bisphenol A and phthalates, may lead to inflammation-driven behavioral changes reminiscent of ASD [84,85,86,87,88]. This action may be through immune and epigenetic mechanisms, or as endocrine disrupters. In this regard, plastic food containers and films were tested and found to contain thousands of chemicals, with many being endocrine disruptors (including an estrogen receptor alpha activator and an androgen receptor inhibitor) and metabolism disruptors [89]. Samples tested that were made with fewer chemicals also had fewer toxic chemicals that entered the food. Additionally, exposure to polystyrene nanoplastics caused depression and anxiety in a mouse model [90], and it decreased oligodendrocyte activity and enhanced hyperactivity and aggression in zebrafish [91], while exposure to polyethylene led to repetitive behaviors and diminished social interaction [92].

#### 3.2.5. Malnutrition

Deficiencies in nutrients with anti-inflammatory properties, such as omega-3 fatty acids, zinc, and vitamins B12, B9, A, D, and K, may increase susceptibility to inflammation and neurodevelopmental disorders [93,94,95,96,97,98]. The vitamin A deficiency may have its effect in an impact on the gut microbiome [99,100]. Other micronutrients appear to play important roles in neurotypical development [101]. There is also evidence that autistic children, who frequently eat the same food each day, are lacking in certain nutrients [102,103], or are receiving an abundance of one macronutrient to the exclusion of others, which may contribute to symptoms [104].

#### 3.2.6. Emotional Stress

Maternal stress during pregnancy can result in elevated cortisol and pro-inflammatory cytokines, potentially disrupting fetal brain development. Chronic stress in the early life of the child may also affect immune and neurological outcomes [57,62,105,106,107,108]. One measure of such stress on the child can be quantified by evaluating adverse childhood experiences (ACEs), as ASD children, who are at increased risk of bullying, parental divorce, income, and food insecurity, have an increased risk of comorbid health disorders [109].

#### 3.2.7. Pharmaceuticals

Certain medications, particularly selective serotonin reuptake inhibitors (SSRIs), have been implicated in altering fetal brain development when used during pregnancy [110]. SSRIs cross the placental barrier and can influence serotonin signaling pathways critical to neurodevelopment. Research suggests that prenatal SSRI exposure may disrupt synaptogenesis and increase neuroinflammation, particularly in genetically susceptible offspring [111]. These effects may contribute to ASD-like phenotypes in animal models and warrant a careful consideration of risk–benefit ratios in clinical practice [110,112,113,114].

#### 3.2.8. Maternal Disorders and Diabetes

A recent study in California found a higher ASD incidence in children born to mothers with asthma or obesity, and a much higher incidence for mothers with both conditions [115]. The authors speculate that this incidence increase may be due to the increased inflammatory conditions during pregnancy, and that the earlier screening of children born to mothers with asthma and obesity during pregnancy may be warranted.

A meta-analysis of over 200 studies, and including over 50 million pregnancies, revealed that children born to mothers with diabetes have an increased risk of neurodevelopmental disorders. This was especially true for mothers with pre-gestational diabetes, and less so for gestational diabetes. There was a 25–30% increased risk for ASD, ADHD, and intellectual disability, with a 15–20% increased risk for communication learning disorders [116].

#### 3.2.9. Synthetic Vitamins 

Folic acid supplementation has been used in prenatal vitamins for decades and reduces spina bifida in newborns. Folic acid is used rather than natural folate due to its stability; it cannot be further oxidized. However, excessive or poorly metabolized synthetic folic acid may provoke immune responses, especially in individuals with impaired folate pathways. This imbalance can exacerbate inflammation and contribute to FRAA development, which may lead to CFD. Additionally, the intake of folic acid in excess of a few hundred micrograms per day (the typical level of folic acid that can be converted to usable folate in the gut) has been shown to result in unmetabolized folic acid in the blood, which interferes with the absorption of folate across the blood–brain barrier or across the placenta [117,118,119,120].

#### 3.2.10. Vaccines

While the scientific consensus supports vaccine safety, there is speculation that vaccines trigger inflammatory responses in genetically susceptible individuals, as detailed in a recent comprehensive report [121]. If such inflammation occurs during critical periods, it can elevate the risk of altering brain development. This was a larger concern in previous decades, for example, when the earlier pertussis vaccine (in the 1980s) triggered fever in some children, or when the thiomersal (thimerosal) preservative in the measles vaccine was used (until 2001), which could be inflammatory in some children. The fact that diagnosed autism cases have increased five-fold from 2000 to 2025 [2,122] while inflammation from vaccines has diminished indicates that this is likely not a significant factor in this neurodevelopmental disorder.

#### 3.2.11. Microbiome and Metabolic Disorders

This topic may merit a separate category, as processing by the microbiome seems to be essential for the absorption of key nutrients as well as the inactivation of inflammatory agents, and digestive disorders are frequently comorbid with ASD [122,123,124,125]. In this regard, restoring microbiome function with probiotics may alleviate some ASD symptoms [126], and the use of fecal transfer has shown some effectiveness in treating autism symptoms [127].

Several studies have shown a link between the gut microbiome and neurological disorders [128]. Propionic acid, a microbial byproduct, is known to cause neuroinflammation and the over-proliferation of glia in mice offspring born to mothers exposed to propionic acid throughout pregnancy and weaning [129]. This provides direction as to a mechanism of action by which inflammation may be caused, as well as neural correlates of its action.

A screening of the gut microbiome of autistic children and their non-autistic siblings revealed categories of bacteria and fungi that were at different levels in the ASD individuals [130]. When these bacteria and fungi were fed to mice, there was a concomitant change in their behavioral patterns, indicating that the use of certain probiotics may provide some benefit in autism. A similar study found differences in ASD and non-ASD microbiomes, and observed that regressed ASD children had higher levels of *Proteobacteria* [131]. This link to ASD regression may provide another avenue to explore.

An additional evaluation of mitochondrial metabolism found changes in the energy pathways in ASD children [132]. When a metabolic analysis was made of cord blood to look at such parameters at birth, as well as in maternal mid-gestational blood, a machine-learning analysis found that cord blood was more predictive of ASD and that this was true for girls and boys [133]. A later study showed that the elevation of a specific cord blood component, acylcarnitine, was predictive of both ASD and ADHD outcomes [134].

### 3.3. Mechanistic Insights 

Inflammation can alter neural circuitry by affecting synaptic pruning, neuronal connectivity, and neurotransmitter systems, and this may occur due to alterations in the neural immune system. Genetic predispositions, such as those affecting folate metabolism, may intensify the brain’s vulnerability to these inflammatory insults. Microglial priming and sustained cytokine exposure during development may create a neuroimmune environment conducive to ASD [60,135,136,137,138,139].

### 3.4. Therapeutic Interventions

Therapeutic intervention to reduce neuroinflammation typically involves reducing exposure to its causative agents. Given the great diversity of such potential sources, and the rampant exposure in daily living to most of these agents, such action is likely to be futile. Some actions can be clearly taken, such as reducing exposure to the agents that one has control over and seeking medical attention to reduce the risk from illnesses during pregnancy.

Additionally, where mechanisms are posited for inflammatory action, and when the impact of such action can be ameliorated, then the use of specific products may provide benefits. In this regard, with maternal infection, there is a concomitant reduction in N-acetyl cysteine and taurine, associated with neuroinflammation, due to a decrease in microglial cells. Each of these compounds can be supplemented into one’s diet to reduce their impact on the brain’s immune function to deter fetal injury.

There was a recent report in a mouse model of autism that an extract of basil, *Ocimum basilicum*, reduced damage in the pups that were induced by maternal separation and reduced autism behaviors via the antioxidant and anti-inflammatory properties of the extract [140]. This opens an avenue to considering other proven antioxidants in an ASD model. In this regard, the comparison with cancer may be apt, as there are decades of research identifying the mechanisms of action of antioxidant nutritive extracts on cancer cell cultures [5,141]. Some of these may also prove to have an impact in deferring ASD development.

## 4. Integrative Perspective: Gene–Environment Interactions in ASD

### 4.1. Synergistic Effects

Genetic susceptibilities, such as impaired folate transport (described above), may amplify the effects of environmental insults. For example, exposure to synthetic folic acid or infections in a child with FOLR1 or RFC polymorphisms may result in a greater risk of ASD due to insufficient levels of natural folate reaching the brain or the uterus [97,120,139].

### 4.2. Epigenetic Modifications

Disruptions in folate metabolism may lead to altered DNA methylation and gene expression, affecting neurodevelopmental outcomes. These epigenetic changes may bridge the gap between genetic and environmental risk factors.

### 4.3. Implications for Prevention and Treatment 

Screening for CFD in the fetus by testing for parental FRAAs, nutritional support (including prenatal vitamins with natural folate instead of folic acid), and inflammation-reducing strategies during pregnancy could reduce ASD risk [43].

The use of machine learning systems to successfully identify ASD in children could provide much earlier diagnosis [142,143]. Such machine learning provides hope that a similar system may be devised that can identify future ASD risk during pregnancy.

Personalized approaches considering both genetic background and environmental exposures hold promise for future interventions. Follow-up on care in early childhood may successfully identify developing ASD risks and reduce autism severity by treating CFD and nutritional deficits at earlier ages, as well as by reducing inflammatory stimuli and reducing widespread neural inflammation.

We project that ASD treatment may parallel other multifactorial disorders, such as cancer, with a targeted approach to lessen risk and improve neurotypical outcomes, with a similar goal of extended remission to provide an outcome of raised qualify of life. Given that such treatment has a lower societal and health maintenance cost and has been demonstrated to have no deleterious effects, we propose engaging such a model of treatment now.

Figure 3 provides a review of some of the oxidative stress and genetic factors that may give rise to specific symptoms observed in ASD [43].

## 5. Conclusions

ASD arises from a complex interplay between genetic and environmental factors. The genetic variants impairing folate metabolism and inflammatory exposures during critical periods each play significant roles in disrupting brain development. Understanding and addressing these risk factors through early detection, targeted interventions, and public health strategies may help reduce the incidence and severity of ASD in future generations.

Identifying the genetic causes and treating their ramifications, such as treating CFD with levo-leucovorin (the prescribable, pharmacological name of L-folinic acid, which is a natural, reduced, form of folate), has been shown in multiple clinical trials to make a substantial difference in resolving the communication difficulties of a young autistic child [46,47,48]. It is time to have this become the standard of practice for all ASD children at the earliest ages.

Addressing inflammation causes can also be accomplished now, focusing on the most readily addressable ones first. In this way, reducing infections in pregnancy, eliminating the malnutrition of mothers and children, and finding ways to alleviate stress may have a lasting impact in reducing the severity of autism. In addition, providing support during inflammation to help retain immune functioning, such as has been reported by using N-acetyl cysteine and taurine to replenish their levels when depleted by inflammation, may be an appropriate next step. It is most important to address the readily resolvable inflammation causes, specifically stress, malnutrition, and health protection from disease, now, and to extend efforts to work as a society to reduce the impact of inflammatory agents such as microplastics and pollution in order to provide the healthiest environment for the neurotypical brain development in all children, thus building a better and more productive future for all.

## Figures and Tables

**Figure 1 ijms-26-06483-f001:**
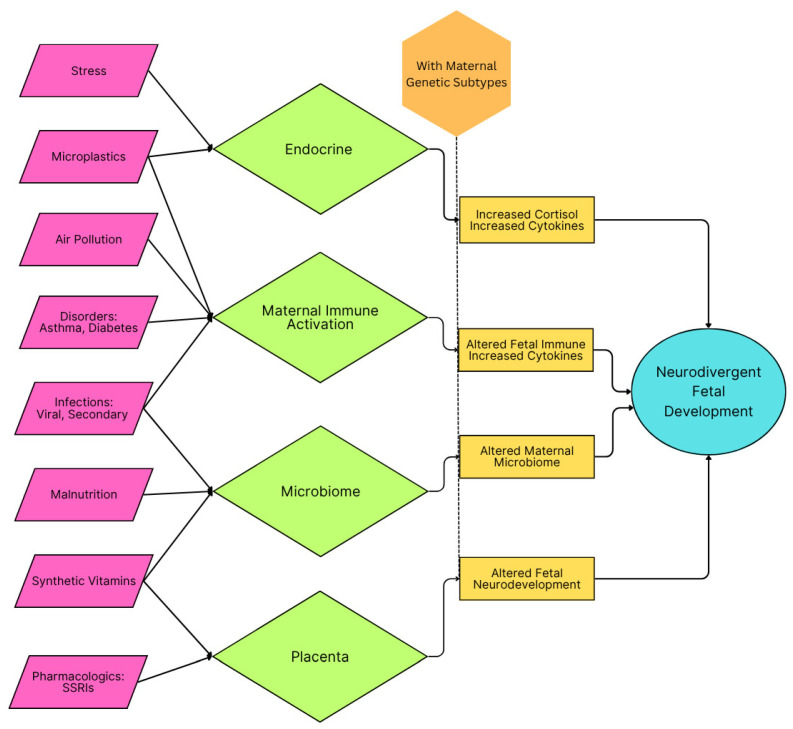
Schematic of maternal inflammatory categories and their impact on fetal developments that can lead to conditions conducive for the development of autism spectrum disorder. Genetic propensity and agents causing oxidative stress are presumed to be the preconditions for neurodivergent fetal development. A similar schematic is seen in Figure 2 for the child.

**Figure 2 ijms-26-06483-f002:**
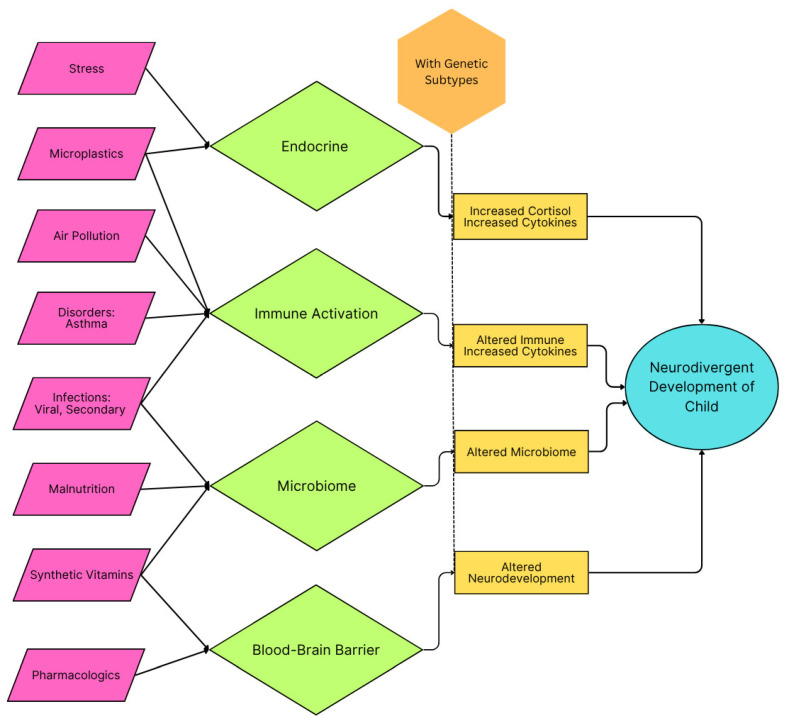
Schematic of inflammatory categories in childhood and their impact on early childhood developments that can lead to autism spectrum disorder symptoms. Genetic propensity and agents causing oxidative stress are presumed to be the preconditions for neurodivergent development.

**Figure 3 ijms-26-06483-f003:**
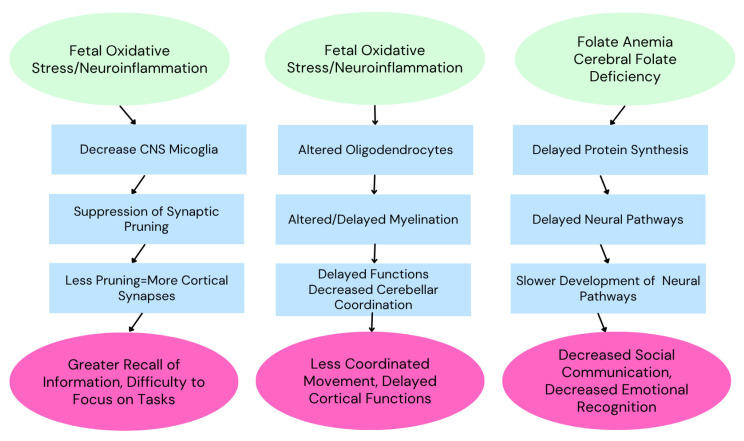
The development of autism spectrum disorder is multifactorial, with factors related to oxidative stress presumably triggering the various symptoms observed. Adapted from Ayoub (2024) [43].

**Table 1 ijms-26-06483-t001:** Commonly referenced genes in SFARI, AutDB, and ClinVar, categorized by area of developmental impact [18].

Category	Genes	Developmental Impact
Synaptic	*ADNP*, *UBE3A*, *GABRB3*, *MECP2*, *NRXN1*, *SHANK3*, *GRIN2B*	Cell Junction Organization
	*ADNP*, *UBE3A*, *GABRB3*, *MECP2*, *NRXN1*, *SHANK3*, *GRIN2B*	Synapse Organization
	*ADNP*, *STXBP1*, *GABRB3*, *MECP2*, *NRXN1*, *SHANK3*, *GRIN2B*	Chemical Synaptic Transmission
	*ADNP*, *GABRB3*, *MECP2*, *NRXN1*, *SHANK3*	Synapse Assembly
Social/ Behavioral	*CHD8*, *MECP2*, *NRXN1*, *SHANK3*	Biological Processes in Intraspecies Interaction
	*CHD8*, *MECP2*, *NRXN1*, *SHANK3*	Social Behavior
Neuronal/ Cellular	*TRIO*, *ADNP*, *UBE3A*, *STXBP1*, *AUTS2*, *MECP2*, *NRXN1*, *TCF4*, *SHANK3*	Neuron Differentiation
	*TRIO*, *ADNP*, *UBE3A*, *STXBP1*, *AUTS2*, *MECP2*, *NRXN1*, *SHANK3*	Neuron Projection Development
	*TRIO*, *ADNP*, *UBE3A*, *STXBP1*, *AUTS2*, *NRXN1*, *SHANK3*	Cell Morphogenesis in Differentiation
	*TRIO*, *ADNP*, *UBE3A*, *STXBP1*, *AUTS2*, *NRXN1*, *SHANK3*	Cell Part Morphogenesis

**Table 2 ijms-26-06483-t002:** Compilation of references linking specific genes with pathways in ASD.

Category	Gene	Pathways and References
Neuronal/ Cellular	*MAPK1*	MAPK signaling, Calcium signaling [23]
	*MAPK3*	MAPK signaling, Calcium signaling [23]
	*HRAS*	MAPK signaling, Calcium signaling [23]
	*PRKCB*	Calcium signaling, MAPK signaling [23]
	*BRAF*	MAPK signaling, Calcium signaling [23]
	*CORO1A*	Neuron function, Immune response [4]
Synaptic	*SCN2A*	Synaptic development (M16 module) [24]
	*SHANK2*	Synaptic development (M16 module) [24]
	*NRXN1*	Synaptic development (M16 module) [24]
	*GRIN2B*	Synaptic transmission [4]
Cellular/ Metabolic	*CNDP1*	mTOR pathway (neurodevelopment) [25]
	*PDE4D*	mTOR pathway (neurodevelopment) [25]
	*ULK2*	mTOR pathway (neurodevelopment) [25]
	*CHD8*	Gene expression in development [4]
	*PTEN*	Cellular development [4]

**Table 3 ijms-26-06483-t003:** SFARI-referenced genes that may be involved in commonly identified ASD comorbidities [10,37,38].

Comorbid Condition	SFARI ASD Gene
ADHD	*PPP3CB*, *PRKG1*
Anxiety Disorder	*ADCYAP1R1*, *DLGAP4*, *NPPB*, *BRP1*, *VIPR2*
Bipolar Disorder	*ADAM10*, *ADCY9*, *ADCYAP1R1*, *AKT1*, *DLGAP4*, *HSPA1L*, *MEGF10*, *NDE1*, *NPPB*, *BRP1*, *VIPR2*
Depressive Disorder	*ADCY9*, *AKT1*, *DGCR8*, *HSPA1L*, *VIPR2*
Epilepsy	*ADCY9*, *AKT1*, *ATN1*, *KCNH2*, *MMP2*, *NDE1*, *SLC29A2*, *SMARCA2*, *VIPR2*
OCD	*ADCYAP1R1*, *DLGAP4*, *NPPB*, *NRP1*, *BIPR2*
Panic Disorder	*ADCYAP1R1*, *DLGAP4*, *NPPB*, *NRP1*, *BIPR2*
Schizophrenia	*ADCY9*, *AKT1*, *ATN1*, *DGCR8*, *DLGAP4*, *HSPA1L*, *KCNH2*, *MEGF10*, *NDE1*, *PPP3CB*, *PRKG1*, *SMARCA2*, *VIPR2*
Sleep Disorders	*ADAM10*, *ADCY9*, *ATN1*, *DGCR8*, *DLGAP4*, *KCNH2*, *MEGF10*, *NPPB*, *NRP1*, *PPP3CB*, *PRKG1*, *SLC29A2*, *SMARCA2*

## Data Availability

No new data were created or analyzed in this study. Data sharing is not applicable to this article.

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
