# Peer review of "Autism Spectrum Disorder as a Multifactorial Disorder: The Interplay of Genetic Factors and Inflammation"

_ijms, 2025, doi:10.3390/ijms26136483_

Round 1
Reviewer 1 Report (New Reviewer)
Comments and Suggestions for Authors
Summary
The manuscript presents a review that requires substantial revisions to strengthen its genetic and pathway analysis of autism spectrum disorder (ASD). The authors must systematically reconstruct the content to properly identify and discuss key ASD inferred genes using aggregation databases such as SFARI, Malacard, and OMIM. A comprehensive analysis of shared pathways through gene ontology and KEGG pathway analysis is essential and must be addressed. The review would significantly benefit from adding a dedicated section covering genetic diagnostic approaches such as genetic panels, whole exome sequencing (WES), and FMR1 expansion analysis. Furthermore, the inclusion of a comprehensive summary table presenting inferred genes, common pathways, and properly formatted references according to MDPI guidelines would greatly enhance the manuscript's utility. These fundamental improvements are necessary before the paper can be considered for acceptance.
Abstract
The abstract section needs enhancement to clearly articulate the main findings regarding predominant ASD-associated genes and pathways. It should specify which pathways are most commonly associated with ASD and identify the gene most predominantly linked to the disorder.
The contribution of environmental factors should be explicitly addressed.
The current mention of "particularly those affecting cerebral folate metabolism" in line 12 requires justification within the context of the broader findings.
These additions would provide readers with a clearer understanding of the review's key contributions.
Introduction
While concise, the introduction lacks important elements that would strengthen its foundation. A brief description of ASD as a neurodevelopmental disorder according to DSM-5 criteria should be included, with reference to DOI:10.1186/2040-2392-4-13. The introduction should also explain ASD severity levels and properly introduce the SFARI database, given its importance for aggregating genetic and phenotypic data (reference DOI:10.1242/dmm.005439). These additions would provide necessary context for readers and better frame the subsequent review content.
Section 2
The current presentation in Section 2 requires clarification and expansion. Line 46 mentions "genes" without specifying which ones, creating confusion. The subsection on gene therapy (2.1.4) needs deeper exploration, particularly regarding monogenic forms of ASD. The authors should incorporate discussion of relevant works (DOI:10.3389/fnmol.2022.1043018 and DOI:10.3390/genes12111667) to strengthen this section. Additionally, the review misses an important opportunity to discuss the intricate association between genes involved in the ubiquitination process (such as UBE3A, ARIH2, CUL3, and UBLCP1) and ASD. References to DOI:10.1038/s41598-024-66475-2, DOI:10.1038/s41598-024-51657-9, and DOI:10.3389/fnmol.2018.00448 would provide valuable support for this discussion.
Author Response
Thank you for the opportunity to improve my manuscript following the guidance of the Reviewers. I am uploading the revised manuscript and am sending the annotated ‘change’ copy for your review of the revisions. To fairly address both reviewers requests for substantial change in the genetic analysis, I have added about 3 pages of content in section 2 (the new 2.1 and 2.2 sections). These review the ASD genes and their impact in development, and the genes found in common among comorbid conditions with ASD.
I address each review in more detail below, following each review paragraph. Reviews are indented.
Review 1
Summary
The manuscript presents a review that requires substantial revisions to strengthen its genetic and pathway analysis of autism spectrum disorder (ASD). The authors must systematically reconstruct the content to properly identify and discuss key ASD inferred genes using aggregation databases such as SFARI, Malacard, and OMIM. A comprehensive analysis of shared pathways through gene ontology and KEGG pathway analysis is essential and must be addressed. The review would significantly benefit from adding a dedicated section covering genetic diagnostic approaches such as genetic panels, whole exome sequencing (WES), and FMR1 expansion analysis. Furthermore, the inclusion of a comprehensive summary table presenting inferred genes, common pathways, and properly formatted references according to MDPI guidelines would greatly enhance the manuscript's utility. These fundamental improvements are necessary before the paper can be considered for acceptance.
I have reconstructed the content to discuss key ASD genes, using SFARI as the primary reference, for consistency. This is seen in the new section 2.1, lines 76-122, as well as new section 2.2, lines 123-154.
Abstract
The abstract section needs enhancement to clearly articulate the main findings regarding predominant ASD-associated genes and pathways. It should specify which pathways are most commonly associated with ASD and identify the gene most predominantly linked to the disorder.
The contribution of environmental factors should be explicitly addressed.
The current mention of "particularly those affecting cerebral folate metabolism" in line 12 requires justification within the context of the broader findings.
These additions would provide readers with a clearer understanding of the review's key contributions.
The abstract has been revised, as indicated in the marked copy, as request. In particular, I have attempted to be more clear as to the main findings and focus of this review. I have explicitly named environmental factors, and reduced the ‘particularly those affecting …’ phrase at issue to simply indicate ‘including those affecting …’
Introduction
While concise, the introduction lacks important elements that would strengthen its foundation. A brief description of ASD as a neurodevelopmental disorder according to DSM-5 criteria should be included, with reference to DOI:10.1186/2040-2392-4-13 [Volkmar 2013]. The introduction should also explain ASD severity levels and properly introduce the SFARI database, given its importance for aggregating genetic and phenotypic data (reference DOI:10.1242/dmm.005439). These additions would provide necessary context for readers and better frame the subsequent review content.
The introduction has been rewritten, and starts now with a reminder of ASD identification, and the source for this. I have used the references the reviewer kindly recommended. The introduction now starts with this, so the first paragraph is mostly new, and reviews both the identification and characterization of ASD, as per the DSM-5.
Section 2
The current presentation in Section 2 requires clarification and expansion. Line 46 mentions "genes" without specifying which ones, creating confusion. The subsection on gene therapy (2.1.4) needs deeper exploration, particularly regarding monogenic forms of ASD. The authors should incorporate discussion of relevant works (DOI:10.3389/fnmol.2022.1043018 [Wang 2022] and DOI:10.3390/genes12111667 [Weuring 2021]) to strengthen this section. Additionally, the review misses an important opportunity to discuss the intricate association between genes involved in the ubiquitination process (such as UBE3A, ARIH2, CUL3, and UBLCP1) and ASD. References to DOI:10.1038/s41598-024-66475-2 [Vinci 2024], DOI:10.1038/s41598-024-51657-9 [Tener 2024], and DOI:10.3389/fnmol.2018.00448 [Vatsa 2018] would provide valuable support for this discussion.
This is the section that has been substantially revised. Section 2 starts at line 55 now, and from line 55-155 it is new or substantially rewritten. This encompasses the new sections 2.1 and 2.2 (displacing the previous such sections to 2.3 and 2.4). The references kindly provided by the Reviewer are used, there is a paragraph on ubiquitination (lines 94-101), there is more specificity on monogenic versions in the opening paragraphs of section 2, and I’ve further added more content on the categorization of ASD genes (section 2.1) and common genes identified in comorbid conditions (section 2.2).
Reviewer 2 Report (New Reviewer)
Comments and Suggestions for Authors
The author reviews autism as a multifactorial disorder in respect to genetic and inflammatory aspects.
The author had already published two recent reviews with similar focus and now bringing genetic factors into account. The genetic part is in general underrepresented making the claimed correlation between genetic factors and inflammation not strong. The author could improve this by not only mention that 70 genes are directly connected to autism but making visible e.g. in a table or in a string database network and pointing those genes/pathways out that are (in)direct connected to inflammation. In addition the author should state if he focusses on isolated autism or autism as a feature of an underlying (syndromal) disease, which for sure is different.
The introduction sounds somehow dubious when the author compares autism with cancer and states ‘…For example, diets that reduce inflammation or support detoxification pathways are known to reduce cancer risk and may contribute to better outcomes in ASD as well.’ without providing any reference!
The author states that one main aspect between nutrition/inflammation and autism is folate supplementation, which in first line is one critical factor also in the pre prenatal phase to avoid neural tube defects or heart defects. Therefore, gynecologists recommend folate substitution to every woman that wants to become pregnant. Due to the argumentation of the author I wonder if there are already some studies that address this also with autism risk or should at least be commented as the author later in the text (paragraph 3.2.9) also gives some contraindications.
The conclusion section again sounds dubious when the author recommends certain drugs for therapy, without outlining in the manuscript before, providing references or distinguishing between isolated and syndromal autism.
Besides those main points, there are some minor points to be addressed:
- according to nomenclature gene names need to be written italic
- Figure 1 and two are similar and could be merged
- Figure 3 provides not really a ‘multifactorial’ overview on factors as genetic factors are missing
- Abbreviation list should be prepared in alphabetic order
Author Response
Thank you for the opportunity to improve my manuscript following the guidance of the Reviewers. I am uploading the revised manuscript and am sending the annotated ‘change’ copy for your review of the revisions. To fairly address both reviewers requests for substantial change in the genetic analysis, I have added about 3 pages of content in section 2 (the new 2.1 and 2.2 sections). These review the ASD genes and their impact in development, and the genes found in common among comorbid conditions with ASD.
I address each review in more detail below, following each review paragraph. Reviews are indented.
Review 2
The author reviews autism as a multifactorial disorder in respect to genetic and inflammatory aspects.
The author had already published two recent reviews with similar focus and now bringing genetic factors into account. The genetic part is in general underrepresented making the claimed correlation between genetic factors and inflammation not strong. The author could improve this by not only mention that 70 genes are directly connected to autism but making visible e.g. in a table or in a string database network and pointing those genes/pathways out that are (in)direct connected to inflammation. In addition the author should state if he focusses on isolated autism or autism as a feature of an underlying (syndromal) disease, which for sure is different.
I have now specifically discussed the number of genes and databases of these in the first paragraph of section 2, where I also explain my focus on non-syndromal autism in this review. The helpful suggestion of a table was useful, and I’ve incorporated a few new tables (Table 1, 2, and 3) to try to provide a visual guide for readers to begin to categorize the genes involved.
The introduction sounds somehow dubious when the author compares autism with cancer and states ‘…For example, diets that reduce inflammation or support detoxification pathways are known to reduce cancer risk and may contribute to better outcomes in ASD as well.’ without providing any reference!
I apologize for omitting any cancer/nutrition reference in the original submission and have rectified it in this revision, and I also included a reference to depression and nutrition in the same paragraph to help show a continuity of potential treatment venues. These are in lines 39-44.
The author states that one main aspect between nutrition/inflammation and autism is folate supplementation, which in first line is one critical factor also in the pre prenatal phase to avoid neural tube defects or heart defects. Therefore, gynecologists recommend folate substitution to every woman that wants to become pregnant. Due to the argumentation of the author I wonder if there are already some studies that address this also with autism risk or should at least be commented as the author later in the text (paragraph 3.2.9) also gives some contraindications.
This is a very important point, and I thank the Reviewer for alerting me that it was unclear. I have revised section 3.2.9 (lines 285-295) in order to clarify that folic acid currently used is oxidized, while the form of folate the body uses is reduced. I discuss folic acid in excess of the amount converted by the microbiome each day interferes with natural folate absorption. I hope this is clear for readers now.
The conclusion section again sounds dubious when the author recommends certain drugs for therapy, without outlining in the manuscript before, providing references or distinguishing between isolated and syndromal autism.
I revised this part of the conclusion to include the references to the clinical trials of the prescription vitamins in treatment, and point out that while it is classified as a drug product, it is a natural vitamin (there are historical reasons for the classification, as it was used in cancer therapy to replace depleted folate for many decades, and thus used in high levels in an IV drip).
Besides those main points, there are some minor points to be addressed:
- according to nomenclature gene names need to be written italic
- Figure 1 and two are similar and could be merged
- Figure 3 provides not really a ‘multifactorial’ overview on factors as genetic factors are missing
- Abbreviation list should be prepared in alphabetic order
I have corrected gene names to all be in italic. I wanted to merge figs 1 and 2, but it makes a rather confusing single figure due to needing multiple options depending on whether it is mother or child. I have retained both figures in the interest of clarity. I have alphabetized the abbreviation list.
Round 2
Reviewer 1 Report (New Reviewer)
Comments and Suggestions for Authors
Authros addressed all the reviewer's comments. Abstract must be reinforced emphasizing on the most important findings of this review.
Reviewer 2 Report (New Reviewer)
Comments and Suggestions for Authors
The author improved the manuscript according to reviewer comments. The manuscript is now ready to be published.
This manuscript is a resubmission of an earlier submission. The following is a list of the peer review reports and author responses from that submission.
Round 1
Reviewer 1 Report
Comments and Suggestions for Authors
This review aims to summarize the interplay between genetic factors and inflammation in Autism Spectrum Disorder (ASD), with particular attention to cerebral folate metabolism and environmental influences. While the topic is important and the manuscript is clearly written, the coverage is relatively narrow and does not provide a sufficiently comprehensive or updated synthesis of the current literature. The genetic discussion focuses almost exclusively on folate-related mechanisms, without addressing the broader landscape of ASD-associated genes identified through genomic studies. As a result, the manuscript may give an imbalanced view of the genetic contributions to ASD.
Several claims in the manuscript would benefit from more cautious framing and stronger evidentiary support. For example, the discussion around vaccines, folic acid supplementation, and the possibility of “remission” in ASD includes speculative elements that are not fully contextualized by established research. These sections would be strengthened by referencing larger epidemiological studies and presenting a more balanced overview of the existing evidence.
In addition, the referencing could be improved. Some key statements lack appropriate citations, and there appear to be duplicate or misnumbered references. A more thorough and critical appraisal of the cited literature, including higher-impact or more recent sources, would enhance the manuscript’s scientific rigor and credibility.
Overall, while the manuscript addresses a relevant area of ASD research, significant revisions would be required to broaden its scope, balance its interpretations, and improve citation practices. I recommend that the manuscript not be accepted in its current form.
Author Response
Response to Review 1:
Thank you for your recommendations. I have gone through and added more content on other genetic components that may be involved, and thoroughly updated the references. The changes are throughout the manuscript, as no section is unchanged. Additionally, I have added a few subsections to address several of the points at issue.
I have also endeavored to temper my comments by using 'may' rather than 'can' or 'does' through the text. I appreciate your caution that the previous wording could be misleading, or could imply that certain topics were conclusive when in fact this is very much a work in progress.
Review 1:
This review aims to summarize the interplay between genetic factors and inflammation in Autism Spectrum Disorder (ASD), with particular attention to cerebral folate metabolism and environmental influences. While the topic is important and the manuscript is clearly written, the coverage is relatively narrow and does not provide a sufficiently comprehensive or updated synthesis of the current literature. The genetic discussion focuses almost exclusively on folate-related mechanisms, without addressing the broader landscape of ASD-associated genes identified through genomic studies. As a result, the manuscript may give an imbalanced view of the genetic contributions to ASD.
Several claims in the manuscript would benefit from more cautious framing and stronger evidentiary support. For example, the discussion around vaccines, folic acid supplementation, and the possibility of “remission” in ASD includes speculative elements that are not fully contextualized by established research. These sections would be strengthened by referencing larger epidemiological studies and presenting a more balanced overview of the existing evidence.
In addition, the referencing could be improved. Some key statements lack appropriate citations, and there appear to be duplicate or misnumbered references. A more thorough and critical appraisal of the cited literature, including higher-impact or more recent sources, would enhance the manuscript’s scientific rigor and credibility.
Overall, while the manuscript addresses a relevant area of ASD research, significant revisions would be required to broaden its scope, balance its interpretations, and improve citation practices. I recommend that the manuscript not be accepted in its current form.
Reviewer 2 Report
Comments and Suggestions for Authors
This is a mini-review reporting a cerebral folate deficiency and a series of different and very heterogeneous factors which could determine inflammation in the autistic brain.
Actually, I do not think that these topics should deserve a review, considering that they have not been confirmed by solid literature data on ASD.
Author Response
Response to Review 2:
Thank you for your recommendation. I have substantially revised the original to take it from a mini-review to a review, which I hope helps address your concern. The referencing is now doubled, and the manuscript has a number of new subsections, which has increased it length and, I trust, its utility in the field.
Review 2:
This is a mini-review reporting a cerebral folate deficiency and a series of different and very heterogeneous factors which could determine inflammation in the autistic brain.
Actually, I do not think that these topics should deserve a review, considering that they have not been confirmed by solid literature data on ASD.
Reviewer 3 Report
Comments and Suggestions for Authors
This paper unifies cerebral folate deficiency (CFD) and neuroinflammation as synergistic drivers of ASD, and highlights folate receptor autoantibodies (FRAAs) as a bridge between genetic susceptibility and environmental triggers (e.g., diet, toxins).
Some references following also focused on inflammatory alterations in ASD. Please review them and update the paper. Some sections (e.g., patients, animal trial) would benefit from recent clinical trial evidence.
Gevezova M, Ivanov Z, Pacheva I, et al. Bioenergetic and Inflammatory Alterations in Regressed and Non-Regressed Patients with Autism Spectrum Disorder. Int J Mol Sci. 2024;25(15):8211. Published 2024 Jul 27. doi:10.3390/ijms25158211
Lagod PP, Abdelli LS, Naser SA. An In Vivo Model of Propionic Acid-Rich Diet-Induced Gliosis and Neuro-Inflammation in Mice (FVB/N-Tg(GFAPGFP)14Mes/J): A Potential Link to Autism Spectrum Disorder. Int J Mol Sci. 2024;25(15):8093. Published 2024 Jul 25. doi:10.3390/ijms25158093
Amini-Khoei H, Taei N, Dehkordi HT, et al. Therapeutic Potential of Ocimum basilicum L. Extract in Alleviating Autistic-Like Behaviors Induced by Maternal Separation Stress in Mice: Role of Neuroinflammation and Oxidative Stress. Phytother Res. 2025;39(1):64-76. doi:10.1002/ptr.8360
Author Response
Response to Review 3:
Thank you for your comments, and for the references you recommended including. I have substantially revised the manuscript with your advice taken through it. The references you cited were very useful, and I've included them and several others in the revision, as I have added some subsections in the process of making the update to the manuscript. I am particularly appreciative to have the references that show other nutritive therapies that proved beneficial. I hope that the revised manuscript may more fully address your points.
Review 3:
This paper unifies cerebral folate deficiency (CFD) and neuroinflammation as synergistic drivers of ASD, and highlights folate receptor autoantibodies (FRAAs) as a bridge between genetic susceptibility and environmental triggers (e.g., diet, toxins).
Some references following also focused on inflammatory alterations in ASD. Please review them and update the paper. Some sections (e.g., patients, animal trial) would benefit from recent clinical trial evidence.
Gevezova M, Ivanov Z, Pacheva I, et al. Bioenergetic and Inflammatory Alterations in Regressed and Non-Regressed Patients with Autism Spectrum Disorder. Int J Mol Sci. 2024;25(15):8211. Published 2024 Jul 27. doi:10.3390/ijms25158211
Lagod PP, Abdelli LS, Naser SA. An In Vivo Model of Propionic Acid-Rich Diet-Induced Gliosis and Neuro-Inflammation in Mice (FVB/N-Tg(GFAPGFP)14Mes/J): A Potential Link to Autism Spectrum Disorder. Int J Mol Sci. 2024;25(15):8093. Published 2024 Jul 25. doi:10.3390/ijms25158093
Amini-Khoei H, Taei N, Dehkordi HT, et al. Therapeutic Potential of Ocimum basilicum L. Extract in Alleviating Autistic-Like Behaviors Induced by Maternal Separation Stress in Mice: Role of Neuroinflammation and Oxidative Stress. Phytother Res. 2025;39(1):64-76. doi:10.1002/ptr.8360
Round 2
Reviewer 1 Report
Comments and Suggestions for Authors
The Author has made significant improvements to this review article.